# Liquid Metal-Based Frequency and Pattern Reconfigurable Yagi Antenna for Pressure Sensing

**DOI:** 10.3390/s25051498

**Published:** 2025-02-28

**Authors:** Xiaofeng Yang, Xiang Ma, Jiayi Yang, Yang Li, Meiping Peng, Qi Zheng

**Affiliations:** 1School of Electronic Engineering, Xi’an University of Posts and Telecommunications, Xi’an 710061, China; yangxiaofeng@xupt.edu.cn; 2School of Information Engineering, Shaoguan University, Shaoguan 512005, China; maxiang.cs@outlook.com; 3College of Computer Science and Technology, Xi’an University of Science and Technology, Xi’an 710054, China; jyang46@xust.edu.cn; 4School of Internet of Things Engineering, Jiangnan University, Wuxi 214122, China; liyang@jiangnan.edu.cn; 5School of Communication and Information Engineering, Shanghai University, Shanghai 200444, China; qizheng@shu.edu.cn

**Keywords:** Yagi antenna, liquid metal switch, frequency- and pattern-reconfigurable, stress sensing

## Abstract

In this work, a frequency- and pattern-reconfigurable Yagi antenna based on liquid metal (LM) switches is proposed for pressure sensing and health monitoring. The proposed antenna consists of a dipole radiator, a reflector, a director, a dielectric substrate, and four flexible LM switches. Benefitted from the switching effect of the LM switches under external pressure, the frequency and radiation pattern of the antenna can be reconfigured. When the LM switch is fully or partially turned on, the radiation directions of the antenna are bidirectionally end-shot and end-fired, respectively. The operating frequency of the antenna can be tuned from 2.28 GHz to 2.5 GHz. It is shown that a maximum gain of 6 dBi can be obtained. A sample was fabricated and measured, and the experimental results were in good agreement with the simulations. The reconfigurable antenna can be applied in wireless pressure-sensing and health-monitoring systems.

## 1. Introduction

Reconfigurable antennas are of great significance for the miniaturization of multifunctional communication systems [1,2,3]. Applications with antenna reconfigurability, including PIN diodes [4,5,6], microelectromechanical system (MEMS) switches [7], and varactors [8], present the advantages of having a small size and a fast switching speed, but they require a bias circuit, which increases the complexity and influences the radiation performance. MEMS switches are advantaged by a low insertion loss and high isolation. However, their low reliability, material fatigue, and high cost limit their application. PIN diodes and MEMS switches are widely used in reconfigurable antennas, but polarization circuits need to be designed for these applications, which increase circuit complexity. Pressure-controlled liquid metal switches are directly controlled by pressure, reducing the complexity of the circuit, making it simple and easy to operate. Developing new reconfigurable antenna mechanisms is of great significance [9,10].

LM has the properties of metal conductivity and liquid fluidity [11,12]. The liquid metal eutectic gallium indium (EGaIn) has low toxicity, high electrical conductivity (3.4 × 10^6^ S/m), and very good flexibility [13], providing new strategies for reconfigurable antennas. In 2018, S. I. H. Shah et al. proposed a frequency-reconfigurable quasi-Yagi antenna consisting of a metal-printed driven dipole and three directors [14]. To tune the frequency and maintain a high gain, microfluidic channels were integrated into radiating dipoles and directors. In 2020, K. Y. Alqurashi et al. proposed a liquid metal bandwidth-reconfigurable antenna that used liquid metal segments to connect or disconnect a large antenna floor [15], which could switch the operating bandwidth between ultra-wideband and narrowband. In 2019, L. Song et al. proposed a frequency-reconfigurable antenna based on liquid metal microfluidic channels [16]. Frequency reconfiguration was achieved by filling the microfluidic channels on the narrow slits with liquid metal, which was controlled either by the pressure difference in the syringe or pump or the voltage difference generated by the circuit. Regarding health monitoring, pressure sensors play an important role in grasping dynamic changes in humans, promoting the application of LM.

In this paper, an LM-based frequency- and pattern-reconfigurable Yagi antenna is proposed for pressure sensing. Due to its advantages of a simple structure [17] and easy fabrication [18], a Yagi antenna is used as the main body of the proposed antenna. The working states are controlled by LM switches without an external bias circuit. The working frequency of the antenna can be reconfigured from 2.28 GHz to 2.5 GHz. The radiation pattern of the antenna can be switched from bidirectional end-shot to end-fire. The feasibility of this reconfigurable antenna design is demonstrated by measurement and simulation. The reconfigurable antenna can also be used to monitor stress [19,20,21], and the stress state of the antenna can be determined by judging the radiation direction and operating frequency, meaning that it could be used as a wireless stress monitor in medical applications.

## 2. Antenna Design and Performance

### 2.1. Antenna Topology

The proposed frequency- and pattern-reconfigurable Yagi antenna works at a center frequency of 2.4 GHz for ISM applications. The radiator is a dipole, and its length is about $\left(0.45\sim 0.48\right)\lambda_{g}$. Reflectors/directors A and B act either as reflectors or directors. The range of the reflector length is generally $\left(0.5\sim 0.55\right)\lambda_{g}$, and that of the director is generally $\left(0.38\sim 0.46\right)\lambda_{g}$. The antenna is based on a substrate made with FR4 dielectric material and a relative permittivity of $\varepsilon_{r}=4.4$. The layout is shown in Figure 1a–c. Switches S1, S2 and S3, S4 are placed on reflectors/directors A and B, with a gap length of B3, respectively. The radiator in this design is mainly dipole, and the liquid metal switch mainly affects the length of the reflector/director. The liquid metal switch does not affect the radiator. The final antenna sizes after optimization analysis are the following: A_1_ = 70 mm; A_2_ = 4.96 mm; A_3_ = 4.7 mm; A_4_ = 7 mm; B_1_ = 65 mm; B_2_ = 11 mm; B_3_ = 5 mm; B_4_ = 30 mm; B_5_ = 20.7 mm; C_1_= 2 mm; C_2_ = 1.5 mm; and H_1_ = 0.8 mm.

### 2.2. LM Switch Design

The flexible switch based on LM fiber benefits from a simple process, low costs, and a strong anti-interference ability. LM fibers are sensitive to stress and can be used not only as switches but also to monitor stress. The LM switch is composed of a hollow fiber and LM [22]. The fabrication process is shown in Figure 2. LM is injected into the fiber using a syringe. Two copper wires covered with silver paste are inserted into the fiber. The silver paste can protect the copper wires from being corroded by the liquid metal. UV glue is used to encapsulate the optical fibers to prevent liquid metal spillage.

The prepared LM switch is shown in Figure 3a,b. The switch is soft, so it can be stretched and bent. The cross-section view of the switch is shown in Figure 3b. Applying stress on the switch can decrease the cross-section area of the fiber, increasing the resistance of the fiber according to Pouillet’s law [23]. Figure 3c presents the plot of stress versus resistance. The resistance increases slowly at the beginning of compression. A sharp increment in resistance occurs after 12 MPa, which indicates the disconnection of the LM. The relationship between resistance and strain is shown in Figure 3d, with the fiber exhibiting a switching effect at 70% strain. When the pressure applied by the switch is less than 45 N, the resistance of the liquid metal switch is approximately zero. When the pressure applied by the switch increases from 45 N to 90 N, the resistance of the liquid metal switch increases from 0 kΩ to 1200 kΩ. When the pressure applied by the switch is greater than 90 N, the resistance increases rapidly, and the switch is turned off. Based on these phenomena, this type of fiber can be used as a switch. The durability of liquid metal switches is good: after tens of thousands of presses and bends, the liquid metal switch can still be controlled, turning it on and off using pressure.

### 2.3. Performance

Figure 4 shows the S_11_ and radiation patterns of the antenna simulation in four different states. The resonant frequencies of states 3 and 4 are around 2.4 GHz, and impedance matching is good. At the same time, state 3 and state 4 present unidirectional end radiation with opposite directions. The resonant frequency of state 1 is 2.5 GHz, and the resonant frequency of state 2 is 2.28 GHz. State 1 and state 2 present bidirectional end radiation.

### 2.4. Parametric Study

The parameters that affect the antenna’s return loss S_11_ and resonance frequency are hereby analyzed. The frequency of the Yagi antenna is the same as the resonant frequency of the dipole; therefore, the values of B_5_ and A_2_ of the radiator have evident effects on the antenna frequency, as shown in Figure 5a,b. When B_5_ = 20.7 mm and A_2_ = 4.96 mm, the antenna center frequency is 2.4 GHz. The length of the reflector/director B_4_ also affects the impedance matching of the antenna. Analysis shows that the antenna’s impedance matching is best when the length of the reflector/director A or B (floor) is 40 mm and the length of the other is 30 mm, as shown in Figure 5c,d.

## 3. Experimental Results and Discussion

To verify the proposed design, the antenna shown in Figure 6 was fabricated and measured. The HFSS simulation software was used to design and optimize the parameters of the Yagi antenna. The S_11_ was measured using KEYSIGHT VNA. The radiation pattern was measured in the Microwave Anechoic Chamber. An electronic pressure gauge was used to apply pressure to the LM switches.

There are four reconfigurable states of the antenna, so the antenna parameters of these four states were tested. The three-dimensional models of the four states of compression are shown in Figure 7. When the pressure applied by the switch is less than 90 N, the switch is conductive. When the applied pressure is greater than 90 N, the switch is turned off. For S1, S2, S3, and S4, when all four switches are under a pressure greater than 90 N, they all turn off, corresponding to state 1. When all four switches are under a pressure lower than 90 N, they are all conductive, corresponding to state 2. When S1 and S2 are under a pressure of less than 90 N, S1 and S2 are conductive, and, when S3 and S4 are under a pressure greater than 90 N, they stop being conductive; these last two scenarios correspond to state 3. When S1 and S2 are under a pressure greater than 90 N, S1 and S2 are turned off, and when S3 and S4 are under a pressure lower than 90 N, they turn on; these last two scenarios correspond to state 4. Figure 8a,b present the measured diagrams of the return loss S_11_ and the normalized radiation patterns of the proposed antenna, under the four states. The measured results of the return loss S_11_ and normalized radiation patterns of the four states of the antenna are in good agreement with the simulations.

The four states of the antenna switches and measurement results are summarized in Table 1 and Table 2. Under state 1, the radiation direction is a bidirectional end-shot. The resonant frequency is 2.5 GHz, and the bandwidth is 2.32–2.7 GHz. For state 2, the radiation direction is a bidirectional end-shot. The resonant frequency is 2.28 GHz, and the bandwidth is 2.17–2.41 GHz. Under state 3, reflector/director B acts as a director, reflector/director A acts as a reflector, and the radiation direction points to −Y. The resonant frequency is 2.4 GHz, and the bandwidth is 2.24–2.59 GHz. For state 4, reflector/director B acts as a reflector, reflector/director A acts as a director, and the radiation direction points to +Y. The resonant frequency is 2.4 GHz, and the bandwidth is 2.24–2.58 GHz. The radiation directions of state 1 and state 2 are both a bidirectional end-shot; however, the resonant frequency differs in the two states. The resonant frequency is 2.5 GHz for state 1 and 2.28 GHz under state 2. Thus, we can determine the state by taking the resonant frequency into consideration. For example, if the reconfigurable antenna’s external transmitter emits a signal with a frequency of 2.2~2.6 GHz, a power of 1 mw, and a step length of 100 MHz, and the reflected power is monitored, when the reflected power of the 2.5 GHz signal is very small and almost equals zero, the state is state 1. When the reflected power of the 2.3 GHz signal is very small and almost equals zero, the state is state 2. Therefore, by combining the frequency and the radiation pattern, we can determine which of the four states the wireless stress monitor is in.

When the antenna is used as a wireless stress monitor, the stress state of the switch can be determined according to the return loss S_11_ and the radiation pattern of the antenna. If an antenna is placed at both external end-fire directions of the wireless stress monitor, when only one of the two external antennas can receive the signal, state 3 or state 4 can be determined, based on the actual radiation direction of the antenna. Also, if one were to monitor the compression state of a certain part of the body and said part were to become congested and swollen, with a pressure greater than 90 N, the radiation direction and resonant frequency corresponding to the antenna would change. Under these conditions, medical staff could clearly monitor the health status of the body part in question.

## 4. Conclusion

This article proposes a frequency- and pattern-reconfigurable antenna based on LM switches, which can determine the frequency, bidirectional end-shot, and unidirectional end-shot using the on or off condition of the switches, sensing external pressure. The center frequency of the antenna can be reconfigured from 2.28 GHz to 2.5 GHz. The proposed pressure-controlled LM switch provides a new mechanism for antenna radiation control and reconfiguration. This antenna can be used as a wireless stress monitor to carry out real-time pressure monitoring, and it could also be applied for human health monitoring.

## Figures and Tables

**Figure 1 sensors-25-01498-f001:**
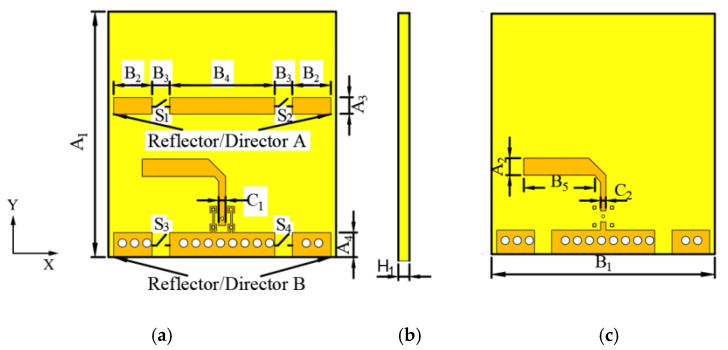
Structure of printed Yagi antenna: (**a**) top view, (**b**) side view, and (**c**) bottom view.

**Figure 2 sensors-25-01498-f002:**
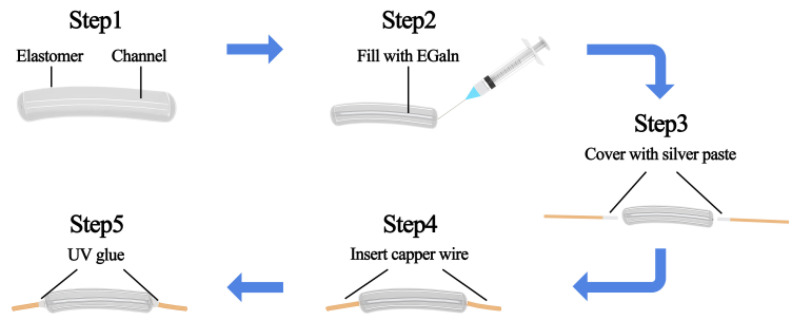
Production flowchart of LM switch.

**Figure 3 sensors-25-01498-f003:**
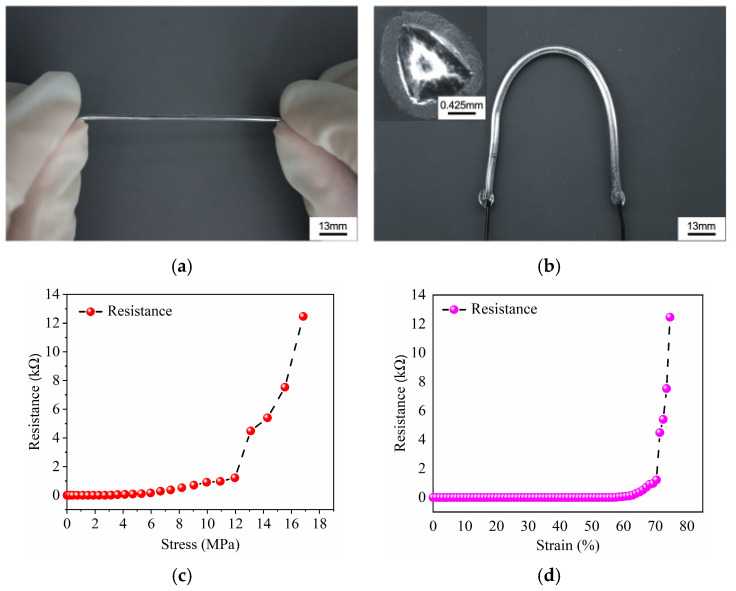
(**a**) Photo of the LM fiber with stretch. (**b**) A bent LM fiber and the cross-section view of the LM fiber. (**c**) Plot of resistance versus stress. (**d**) The resistance of the LM fiber under increasing strain.

**Figure 4 sensors-25-01498-f004:**
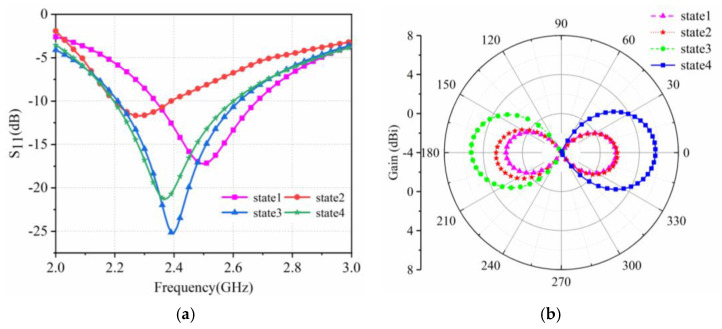
(**a**) Simulated S11 and (**b**) normalized radiation patterns of the proposed antenna under four different states.

**Figure 5 sensors-25-01498-f005:**
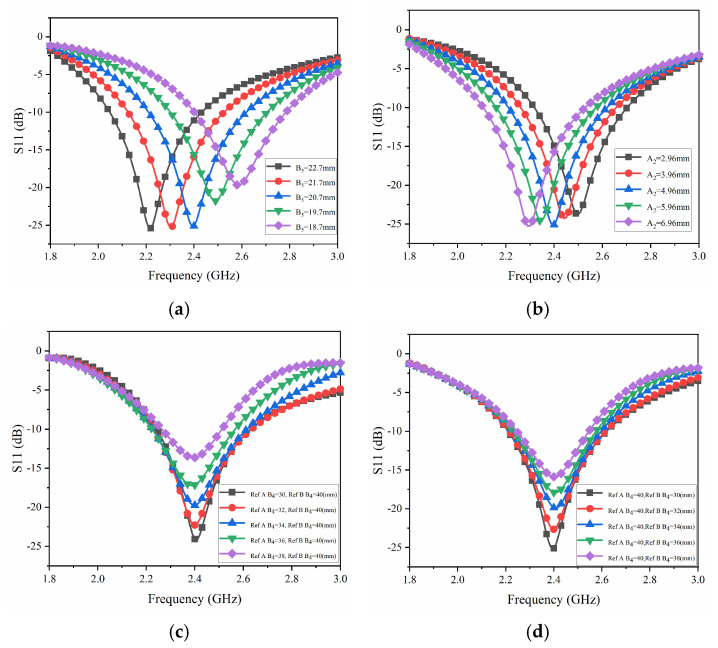
Optimization analysis of antenna S_11_: (**a**) S_11_ with different B_5_, (**b**) S_11_ with different A_2_, and (**c**,**d**) S_11_ with different Ref/Dir A and B.

**Figure 6 sensors-25-01498-f006:**
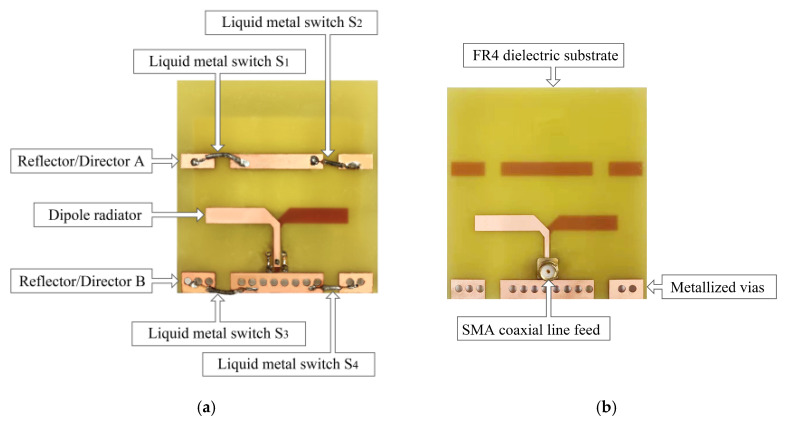
The physical map of the antenna: (**a**) the top and (**b**) the bottom.

**Figure 7 sensors-25-01498-f007:**
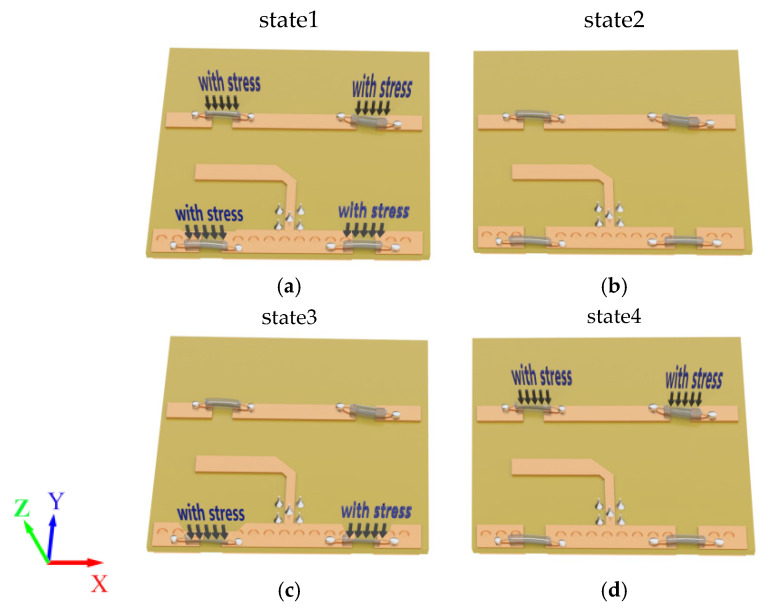
Four stress states of the LM switch: (**a**) state of the four switches with stress; (**b**) state of the four switches without stress; (**c**) state of S3 and S4 switches with stress; and (**d**) state of S1 and S2 with stress.

**Figure 8 sensors-25-01498-f008:**
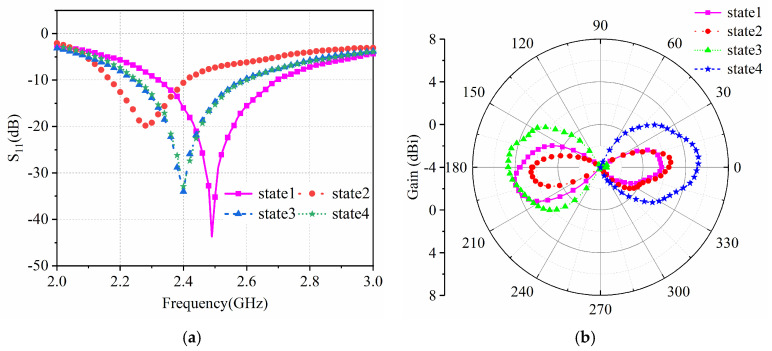
(**a**) Measured S11 and (**b**) normalized radiation patterns of the proposed antenna under four different states.

**Table 1 sensors-25-01498-t001:** The on/off status of the switch corresponding to the four states of the antenna.

State of Antenna	Switch 1	Switch 2	Switch 3	Switch 4	Reflector/Director A		Reflector/Director B	
**State 1**	off	off	off	off				

**State 2**	on	on	on	on				

**State 3**	on	on	off	off	Reflector		Director	
**State 4**	off	off	on	on	Director		Reflector	

**Table 2 sensors-25-01498-t002:** Measured results of four states of the reconfigurable antenna.

Measurement Result	Bandwidth	Resonant Frequency	Gain	Radiation Direction
**State 1**	2.32–2.7 GHz	2.5 GHz	4.1 dBi	Bidirectional end-shot
**State 2**	2.17–2.41 GHz	2.28 GHz	3.6 dBi	Bidirectional end-shot
**State 3**	2.24–2.59 GHz	2.4 GHz	5.6 dBi	Towards −Y
**State 4**	2.24–2.58 GHz	2.4 GHz	6.0 dBi	Towards +Y

## Data Availability

The original contributions presented in this study are included in the article. Further inquiries can be directed to the corresponding author.

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
