# Peer review of "Liquid Metal-Based Frequency and Pattern Reconfigurable Yagi Antenna for Pressure Sensing"

_sensors, 2025, doi:10.3390/s25051498_

Round 1
Reviewer 1 Report
Comments and Suggestions for Authors
Title: Liquid Metal-Based Frequency and Pattern Reconfigurable Yagi Antenna for Pressure Sensing
Manuscript ID: sensors-3390690
In this paper, the author proposes a frequency and pattern reconfigurable Yagi antenna based on liquid metal switches for pressure sensing and health monitoring. The idea of this paper is not new. However, the paper is well organized. Reviewer has some comments below:
1. Provide the far-field distribution of the antenna.
2. Can the author describe the effects of pressure on an antenna using mathematical formulas?
3. The paper lacks a clear explanation of how the LM switches operate under different pressure levels. While the relationship between stress and resistance is mentioned, the detailed mechanism of how this affects the antenna's performance needs to be elaborated.
4. How does the performance of this proposal compare to antennas using PIN diodes or MEMS switches?
5. The paper mentions potential applications in health monitoring and stress sensing but does not provide experimental results for these use cases. Please discuss it in more detail.

The quality of English meets the requirement of Sensors.
Reviewer 2 Report
Comments and Suggestions for Authors
This is interesting work, however, some suggestions are:
1. consider the life time for such switch, how many bending can it last?
2. for automatical operation, will bending manichanisim (not shown in this paper yet) affects the performance of the radiation?
3. how much force is required for stress the switch, will the on/off changes linear to the force applied?
Author Response
Dear editor and reviewer:
Thank you for your email (dated January 24, 2025) in which you attached the reviewers and your valuable comments on our manuscript, titled “Liquid Metal-Based Frequency and Pattern Reconfigurable Yagi Antenna for Pressure Sensing” by Xiaofeng Yang, Xiang Ma, Jiayi Yang, Yang Li, Meiping Peng and Qi Zheng. The authors would like to thank you and the anonymous reviewers for careful reading as well as constructive and thoughtful suggestions. These comments and suggestions have significantly helped the authors to improve the quality of the paper. Moreover, they have also motivated the authors to carefully and deeply think over this research topic. Corresponding modifications have been made in the revised manuscript.
We have carefully revised the manuscript based on the suggestions provided by the reviewers, and the main parts of the revisions are highlighted in red. Below is the authors' detailed responses to the reviewers' suggestions. The reviewers' comments have been highlighted in italic and bold.
Sincerely yours,
Corresponding author: Meiping Peng
School of Information Engineering
Shaoguan University
Shaoguan, CHINA, 512005
- consider the life time for such switch, how many bending can it last?
Reply: Thanks for reminding us. The durability of liquid metal switches is good, after tens of thousands of presses and bends, the liquid metal switch can still be controlled to turn on and off through pressure.
2、for automatical operation, will bending manichanisim (not shown in this paper yet) affects the performance of the radiation?
Reply: Thanks for reminding us. It will not affect. The radiator in this design is mainly dipole, and the liquid metal switch mainly affects the length of the reflector/director. The liquid metal switch will not affect the radiator.
3、how much force is required for stress the switch, will the on/off changes linear to the force applied?
Reply: Thanks for reminding us. When the pressure applied by the switch is in the range of 0N to 90N, the switch changes linearly with the applied force. When the pressure applied by the switch is greater than 90N, the switch resistance increases sharply, and the switch changes nonlinearly with the applied force. The switch state is open.
Reviewer 3 Report
Comments and Suggestions for Authors
Author Response
Please see attachment.

Round 2
Reviewer 1 Report
Comments and Suggestions for Authors
As revised manuscript shown, it has addressed all the relevant points that reviewers required and commented.
It is suggested to accept the manuscript in this form for publication in Sensors (MDPI).